# Rapid handheld time-resolved circularly polarised luminescence photography camera for life and material sciences

Davide F. De Rosa[1], Patrycja Stachelek[1], Dominic J. Black[1] & Robert Pal [1] ✉

Circularly polarised luminescence (CPL) is gaining a rapidly increasing following and finding new applications in both life and material sciences. Spurred by recent instrumental advancements, the development of CPL active chiral emitters is going through a renaissance, especially the design and synthesis of CPL active luminescent lanthanide complexes owing to their unique and robust photophysical properties. They possess superior circularly polarised brightness (CPB) and can encode vital chiral molecular fingerprints in their long-lived emission spectrum. However, their application as embedded CPL emitters in intelligent security inks has not yet been fully exploited. This major bottleneck is purely hardware related: there is currently no suitable compact CPL instrumentation available, and handheld CPL photography remains an uncharted territory. Here we present a solution: an all solid-state small footprint CPL camera with no moving parts to facilitate ad hoc time-resolved enantioselective differential chiral contrast (EDCC) based one-shot CPL photography (CPLP).

Circularly polarised luminescence (CPL), the emission analogue of the widely used technique circular dichroism (CD), is a unique phenomenon that is a hallmark feature of chiral optically emissive molecular systems[1–7]. Just as chirality itself, emission from chiral molecular entities is binary in helicity as left- and right-handed CPL correspond to photons of spin angular momentum $+\hbar/2\pi$ and $-\hbar/2\pi$, respectively[8–10]. Circularly polarised light can be more simply understood by considering a linearly polarised light wave, whose resultant electric-field vector arises from two orthogonal and equal magnitude x and y components of a transverse electromagnetic wave propagating along the z axis[11]; where left and right circular polarization refers to the direction of rotation of the resultant electric- field vector as the wave propagates effectively tracing out a helical pattern like a corkscrew. The hallmark of quarter wave plates (QWPs, λ/4), and in fact all birefringent wave plates, is that they have a fast and a slow axis, orientated at orthogonally (90°) with respect to each other. In the case of left- and right-handed circularly polarised light, wave incident at 45° to the fast axis of a QWP, the phase of the slow axis of the light wave will be retarded by 90° (λ/4), resulting in the wave now becoming vertically and horizontally linearly polarised, respectively[12,13]. In essence this retardation results in the x and y components being in phase so the resultant electric-field vector oscillates in a single plane at 45° to both the x and the y components, resulting in linear polarization.

CPL is quantified in absolute terms by the emission dissymmetry factor ($g_{em}$, Eq. 1)

$$g_{em} = \frac{2(I_{L-CPL} - I_{R-CPL})}{(I_{L-CPL} + I_{R-CPL})}. \tag{1}$$

Where $I_{(L\text{-}CPL)}$ and $I_{(R\text{-}CPL)}$ are the intensities of left-handed (L-CPL) and right-handed (R-CPL) emission; $g_{em} = 2$ indicates complete L-CPL emission, $g_{em} = -2$ indicates complete R-CPL emission, and $g_{em} = 0$ indicates an overall net zero circular polarisation. The strongest CPL signals to-date have been generated by lanthanide(III) coordination complexes, specifically europium complexes, exhibiting $|g_{em}| = 1.38$ for single complexes and $|g_{em}| = 1.45$ for chiral supramolecular polymers[1,14]. Recently, the more comprehensive metric of circularly polarised brightness (CPB) has been introduced[1,15]. CPB serves as a standardised metric for the total number of circularly polarised photons emitted, and combines the molar absorption coefficient

[1]Department of Chemistry, Durham University, South Road, Durham DH1 3LE, UK. ✉e-mail: robert.pal@durham.ac.uk

($\varepsilon_{abs}$), quantum yield ($\phi_{em}$), and $g_{em}$. CPB (Eq. 2, or often denoted as $B_{CPL}$) is calculated as:

$$CPB = \varepsilon_{abs}\, \phi_{em}(g_{em}/2) \qquad (2)$$

It has units of $cm^{-1}\,M^{-1}$. A wide variety of CPL emitting molecular systems can produce useable CPB values (defined here as $\geq 50\ cm^{-1}\,M^{-1}$), including BODIPYs, helicenes, excimers, cyclophanes, and $d$-metal complexes[15]. However, chiral lanthanide complexes are pre-eminent owing to their excellent photophysical properties: large pseudo-Stokes' shifts, narrow, line-like emission spectra, tuneable emission and excitation properties, intricate CPL emission, with the potential for exceptional $g_{em}$ and CPB values (e.g., CPB values of $> 3000\ cm^{-1}\,M^{-1}$ with europium complexes)[2]. CPB is generally maximised for an emission band (or, often, individual transitions thereof) that has a high molar extinction coefficient, high emission quantum yield and a modest $g_{em}$ value. Unfortunately, bands that have high $g_{em}$ values are often not associated with high quantum efficiencies, so there is a trade-off between $g_{em}$ and CPB[15]. Whilst both $g_{em}$ and CPB are very useful metrics, they should not be a blunt comparison tool. Rather, photophysical data and CPL properties of a complex should be considered collectively in the context of the desired application. A complex with a large $g_{em}$ value will maximise contrast for enantioselective CPL imaging[16–18], whereas a complex with high CPB can provide large quantities of circularly polarised photons, ideal for high-throughput rapid CPL verification such as screening banknotes, passports, certificates and IDs.

Triggered by recent milestone developments in CPL instrumentation, the field of CPL spectroscopy and CPL probe design has undergone a renaissance in recent years[6,17–19]. Since the inception of CPL spectrometers over 50 years ago, chiroptical and spectral discrimination has been facilitated using the well-established trinity of photoelastic modulators (PEM), lock-in-amplifiers (LIA) and scanning monochromators (SM) with predominantly custom-built spectrometers[20–22]. The PEM functions as an electronically switchable quarter wave plate (QWP) oscillating at radio frequencies (40–50 kHz), converting L-CPL and R-CPL to orthogonal linearly polarised light that are subsequently analysed by a static linear polariser (LP). The intensity of the resultant radio-frequency oscillating signal is quantified by a single channel photo detector aided by a LIA. Such setups have resulted in bulky, expensive instruments with slow spectral data acquisition rates (-1 h for a typical CPL spectrum)[1,6,23,24]. More importantly, due to the fixed oscillating mechanically induced birefringence resonating nature of the PEM, time-resolved detection of CPL proved to be extremely challenging. The combination of the above has been the major contributing factor for CPL not becoming a widely and routinely utilised spectroscopic technique.

Our newly developed all solid-state (SS-) CPL spectrometer represents a paradigm shift in circularly polarised luminescence spectroscopy[6]. This constitutes the next generation of CPL spectrometers employing radically new instrumental design circumventing all the issues and limitations that hindered the last 50 years of CPL spectroscopy[1], comprising of a patented all-solid-state chiroptical translation unit with a precisely aligned QWP and LP array at its heart. The combination of this, aided by linear CCD detectors, facilitates rapid simultaneous single- or dual-channel time-resolved full spectral detection of left- and right-handed CPL. Several alternative configurations of this chiroptical separator have been tested with the most notable application of a rotatable QWP and LP combination enabling conversion of any conventional fluorescence spectrophotometer or scientific camera into an instrument capable of differential CPL detection (Supplementary Fig. 3)[17,25].

Another prominent area of CPL instrumental development has focused on attempts of translational CPL microscopy. In 2016 we constructed a proof of concept chiroptical contrast time-resolved epifluorescence microscope – equipped with a QWP and a selector of two orthogonally originated LP in the detection path – and demonstrated enantioselective differential chiral contrast (EDCC) imaging. This was achieved by examining the Λ- and Δ-enantiomers of a bright CPL-active europium complex Eu:L4, Eu[1,4,7- tris({4-[2-(4-methoxy-2-methylphenyl)ethynyl]–6-[carboxy(phenyl)phosphoryl]-pyridin-2-yl} methyl)–1,4,7- triazacyclononane]) (EuL:4, structure depicted on Fig. 1B) absorbed into an optical brightener-free paper test substrate[18].

The above-mentioned microscope allowed rapid time-resolved detection. However, it has two drawbacks: single detector channel configuration requiring manual selection and subsequent sequential detection of L-CPL and R-CPL, and low optical sectioning capability due to lack of confocality[26]. Recently, we have demonstrated the extension of our rapid CPL-detection technology into a more desirable laser scanning confocal microscopy (LSCM) set-up[19]. This new approach allows the simultaneous acquisition of left- and right-handed EDCC CPL images with a diffraction-limited spatial resolution (i.e., 126 nm lateral and 396 nm axial resolution using ×63 1.4 NA (numerical aperture) objective) across a typical field of view (FOV) of $100 \times 100\ \mu m$ in 9 s, while employing 355 nm single-photon activation with an Nd:YAG (3rd harmonic) laser using the Λ- and Δ-enantiomers of a bright CPL-active europium complex Eu:L4, Eu[1,4,7-tris({4-[2-(4-methoxy-2-methylphenyl)ethynyl]–6-[carboxy(phenyl)phosphoryl] pyridin-2-yl}methyl)−1,4,7- triazacyclononane]) (Fig. 1) taken up by NIH 3T3 (mouse skin fibroblast, ATCC-CRL-1658) cells. Such microscopes could be used to analyse microscale CPL patterning for security-ink

**Fig. 1 | Chiral lanthanide complexes discussed as candidates for CPL-active security inks. A** Eu:L1, a newly synthesised complex that exhibits desirable strong single sign (exclusively left or right helicity) $\Delta J = 1$ and $\Delta J = 2$ transitions. **B** Eu:L2-5, legacy complexes demonstrating suitable CPL spectra to be used for time-resolved CPLP (TR-CPLP). These complexes have previously been used for both epifluorescence and laser scanning confocal CPL microscopy (see Fig. 2 and Supplementary Fig. 2 for relevant photophysical properties and CPL spectra).

applications. Therefore, in addition to cellular localisation studies, we have imaged several thin and thick film (Supplementary Figs. 10–14) embedded CPL emitters for applications such as physically unclonable multi-layered CPL active security inks[1,6,27].

This research is fuelled by the ever-increasing need for authenticating products and documents with security inks that are vital to global commerce, health, and personal identity documents. Lanthanide complexes are widely used in luminescent security inks as they exhibit many exceptionally unique photophysical properties due to their partially filled $4f$ subshell that is shielded from the ligand environment. The shielding of the $4f$ electrons has a profound effect on the antenna excitation induced optical spectra of lanthanides such as large pseudo-Stokes' shift, long-lived emission, reduced self-quenching, resistance to photobleaching and narrow and line-like emission profiles[28–36]. However, their superior CPL properties, namely the high $g_{em}$ values and high CPB, have not been exploited as an added extra layer of security in advanced security inks. This is largely attributed to the limitations posed by current CPL instrumentation, where rapid hand-held full frame/spectra detection and discrimination of L-CPL and R-CPL are not available. A solution to such instrumental demand is to utilise a full frame/field of view high sensitivity scientific camera aided by our proven simple chiroptical separator unit containing a rotating QWP and precisely aligned stationary LP. The latter is important as due to the unique birefringence of each CCD or CMOS camera chip, rotating the LP as opposed to the QWP will result in light helicity mismatch and randomisation of CPL due to the induced light wave retardance by the detector chip itself. Subsequent spectral and chromatic (colour) selectivity can be achieved by the employment of an appropriate band-pass filter (BPF) to match unique helicity transitions in the emitted light.

Although several CPL-active lanthanide complexes have been developed over the past 50 years, the technology used to measure CPL has been somewhat stagnant, until our recent breakthrough in CPL measurement technology that enables rapid, high-throughput, security-ink verification. A recent study by Kitagawa et al.[17] of enantioselective CPL imaging on a glass substrate used a simple adaptation of our proprietary CPL chiroptical imaging set-up[6] to select for CPL emission a Eu(III) complex deposited onto a glass surface. The complex was generated from the well-known CPL standard {Eu[(+,−)-facam]$_3$} (facam = 3-(trifluoroacetyl) camphorato, a β-diketonato ligand), a compound that was also modified to validate our SS-CPL instrument in 2020[37]. {Eu[(+,−)-facam]$_3$} was mixed with the glass-forming agent TMPO (TMPO = tris(2,6- dimethoxyphenyl)phosphine oxide) and upon excitation with UV light at 365 nm, the individual enantiomers complex exhibited a strong enantiomeric contrast ($g_{em} = \pm1.2$; $\Delta J = 1$) with a total luminescent quantum yield of 13%. They were separately deposited onto a glass substrate and the individual patterns could be clearly differentiated based on left- and right-handed CPL, although an enantioselective contrast ratio was not reported. Alongside our advances in CPL-measurement instrumentation, such as rapid CPL spectroscopy and enantioselective CPL[6,18,19,38], this study is the best example of macroscale enantiomeric contrast imaging to date and clearly demonstrates the potential of advanced CPL active security inks.

We have recently conducted a review explicitly detailing and elaborating on several classes of CPL emitting molecular entities to be used as such security inks[1]. One of the key design parameters for such CPL active lanthanide complexes is to maximise the emission dissymmetry factor ($g_{em}$) and increase CPB with exclusively positive or negative sign CPL across the entire wavelength range of emission bands (electronic transitions) to facilitate ad hoc band-pass filter-based detection. In fact, high dissymmetry factor and high brightness alone are not sufficient to use a CPL-active lanthanide complex as a security ink. The difficulty with lanthanide complexes is that their emission manifolds, labelled by their $\Delta J$ value, are narrow (order of

10 nm) and comprise of multiple transitions. The individual transitions within a single $\Delta J$ manifold of lanthanide complexes often have discordant sign in CPL spectra rendering the overall CPB of the manifold low. This polarisation cancellation effect manifests as a loss of signal even for emitters with a high dissymmetry factor and brightness. However, the goal for a broad diffusion of CPL-based security inks is to have a rapid handheld low-cost and simple instrument to detect CPL. This practical constraint contrasts with the necessity to have bulky, slow, expensive spectrometers to discern opposing sign CPL within each emission manifold. Our proposed solution consists in designing CPL emitters displaying same-sign CPL emission within an individual emission manifold. In terms of detection instrumentation, this allows us to use a bandpass filter to select a single-emission manifold. The selected light will be highly circularly polarised and can be detected using the rapid handheld camera setup proposed in this article. This approach shifts the problem of CPL cancellation effects from the detection instrumentation (the necessity of having a small bandwidth detector to discern opposing sign CPL transitions) to the difficulty of designing lanthanide complexes to finely engineer their CPL spectra, as detailed below.

Spurred on from our advancements in the field of CPL research[1,6,18,19,38], we have coined the term Chameleon Security Inks (CSI) as a new class of intelligent security dyes. These blends of luminescent materials could combine organic short-lived (ns) green/blue fluorophores and chiral (CPL active), high brilliance red/green (europium/terbium) long-lived (ms) emitters embedded into a transparent polymer matrix which are collectively invisible to the naked eye. The chiral molecular fingerprints encoded in the luminescence spectra of CPL active Ln(III) complexes, alongside the possibility of time-resolved colour separation adds two extra layers of security. This allows for multi-layered unclonable QR or bar code generation. They possess an unprecedented five-tiers of security comprises of multi-coloured, multi-spectral, opposing-helicity, combined with high spatial and temporal resolution. The increasing use of polymer-based banknotes allows the introduction of hidden in plain sight security features that combined with the right read-out instrument can facilitate ad hoc verification greatly advancing security and authenticity.

Aside, from the instrumental hurdles to overcome, there are several important physical criteria that need to be mitigated in order to fully utilise luminescent lanthanide complexes as part of intelligent security inks[27]. A key characteristic of Ln(III) security dyes is that they need to be readily soluble to be used as inks and retain required properties on deposition. They must be resistant toward fading caused by photobleaching and be thermally robust to survive conditions such as the relatively high temperatures (often excess of 150 °C for seconds − 1 second per document cm) necessary for document lamination or integration into modern plastic bank notes. Once all the above requirements have been met, the CSIs can easily be embedded into a suitable transparent polymer matrix via simple drop casting, purposeful synthetic engineering, or spin coating.

## Results

Throughout this work when discussing the CPL of Eu(III) complexes, we selected the magnetic dipole (MD) allowed $\Delta J = 1$ transition ($\lambda_{em}$ -595 nm) because it exhibits both high overall emission intensity and a high $g_{em}$ value, resulting in an overall high CPB. Europium complexes with single sign (displaying exclusively left or right helicity) at this MD allowed transition have been identified and used throughout. Therefore, using optical filters, the whole $\Delta J = 1$ manifold can be selected maximising light collection efficiency; it represents the best emission window to examine for Eu(III) complexes (Supplementary Fig. 1). We have demonstrated that the hypersensitive $\Delta J = 2$ transition ($\lambda_{em}$ -610 nm) can also be used for ad hoc CPL discrimination, using a newly synthesised Eu(III) complex, Eu:L1 (Figs. 1, 2), where the complex has

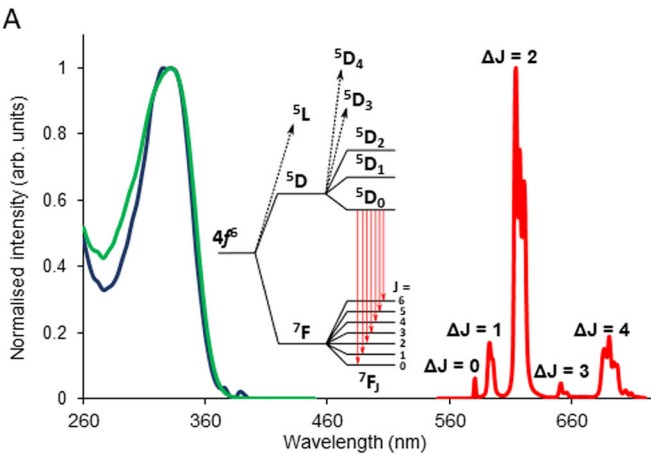

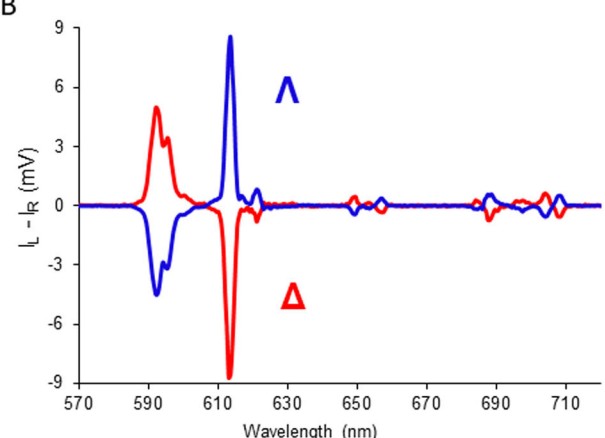

**Fig. 2 | Photophysical characteristics of Eu:L1. A** Absorption (green), excitation (blue, $\lambda_{em}$ = 614 nm) and total emission spectra (red, $\lambda_{ex}$ = 355 nm) of Eu:L1 (c = 5.4 × 10$^{-5}$ M, structure depicted on Fig. 1A) in EtOAc. **B** CPL emission spectra of

Δ- (red) and Λ- (blue) enantiomers of Eu:L1. Both spectra were recorded in EtOAc, 295 K, 5 μM complex, $\lambda_{exc}$ = 355 nm, 5 scans averaged with 0.5 nm resolution and 1 ms integration time.

purposefully been engineered to have opposing sign for $\Delta J = 1$ and $\Delta J = 2$ transitions with high CPB.

### Engineering a bright exclusively positive or negative signed CPL emitter, Eu:L1

As highlighted above, one of the most desirable features of a suitable Eu(III) complex for successful use in combination with a handheld CPL photographic (CPLP) camera is the ability to exhibit single sign (positive or negative), broad emission bands with high CPB. Having exclusively positive or negative sign of CPL within a single manifold improves signal detection due to the absence of sign cancellation, subsequently increasing the number of CPL photons detected. The sign of CPL bands within the $\Delta J = 1$ and 2 manifolds depends on the nature of the donor atoms used to chelate the central lanthanide ion and on the polarity of the environment (see Supplementary Fig. 8). Harnessing our accumulated expertise in the design and synthesis of luminescent lanthanide complexes[27,30,37,39,40], we designed and synthesised a compound displaying the desired single sign properties in both $\Delta J = 1$ and 2 transitions of the CPL emission in low polarity environments. The Eu(III) complex, Eu:L1 (Fig. 1A), bearing a methyl ester functionality and mixed carboxylate and phosphinate donors was synthesised and chirally resolved with complete photophysical characterisation to calculate transition/manifold specific CPB values (Fig. 2, Supplementary Figs. 2, 5 and 7 and Supplementary Tables 1, 2).

Recently we have published the first multiphoton (MP) activated CPL spectrum of lanthanide complexes[19], and determined their all-important two photon excitation (2PE) cross section[41]. One of the great advantages of arylalkynylpyridyl based complexes, such as Eu:L1-5, is that they may be efficiently sensitised via biologically favourable multiphoton – 2PE at $\lambda_{ex}$ = 680–720 nm. We have determined the 2PE cross section of Eu:L1 according to established procedures[41,42] to be $\sigma^2 = 65 \pm 3$ GM (1 GM = 10$^{-50}$ cm$^4$ s photon$^{-1}$, Supplementary Fig. 6.) for both the Λ- and Δ-enantiomers, with complete CPL spectrum recovery recorded upon MP excitation at 680 nm that can also aid achieving another milestone in CPL research MP-CPL-LSCM.

While developing our CPL-LSCM we have reported the feasibility of encapsulating bright, high CPB enantiopure europium complexes within a uniform polymer matrix (PVP-40)[19]. Herein, we demonstrate that Eu:L1 can be encapsulated in the more widely used poly(methyl methacrylate) (PMMA, $M_W$ = 15,000 g mol$^{-1}$). We have employed two different techniques to achieve this: spin coating and drop casting onto a glass surface. The solutions were prepared by blending PMMA with enantiopure solutions of Δ- and Λ-EuL:1 (5 × 10$^{-5}$ M, MeOH) as previously reported[19]. Using these luminescent PMMA thin films

(Supplementary Figs. 6 and 16–18), complete CPL spectrum recovery has been achieved for both enantiomers and the CPL fingerprint was fully preserved after MP excitation (Supplementary Figs. 5–7).

### Compact hand-held enantioselective CPL Photography

As reiterated throughout this publication having a high CPB lanthanide(III) complex encapsulated in a thin film can only truly become part of an ultra-secure anticounterfeit ink if an equally suitable handheld CPL readout device is developed and validated. For most authentication purposes, just as with current invisible security inks, simple chromatic (colour) and EDCC-based ad hoc differentiation is sufficient, eliminating the need for complex spectrographic techniques. Additionally, as a further, next-stage measure of security, off-site high spectral resolution spectroscopic validation of the ink composition and deposition pattern mapping would be available.

To satisfy our compact instrumental design criteria, we have translated our chiroptical separator unit into a solid state, small foot-print single snapshot CPL photographic camera. One of the biggest challenges associated with using the above mentioned QWP and LP array is that it requires precise, reproducible alignment of the QWP to two orthogonally established positions. Despite the availability of precise motorised rotational mounts[6], it is still a critical source of potential systematic error. A way to eliminate such error is to incorporate a scientific (CCD or CMOS array) camera where individual pixel alignment with respect to linear polarisation orientation is achieved resulting in a solid-state linear polarisation sensitive camera. With recent advancements in camera technology, largely driven by detecting stress induced polarisation changes in reflected or transmitted light in material sciences, as such polarisation sensitive scientific cameras have been made available but never been used in the field of CPL or time resolved photography. At the heart of such a camera (a Kiralux® CS505MUP1, ThorLabs) each cluster of repeating four pixels in the camera's 5.0 megapixel array (2448 x 2048 pixels) is masked with a wire grid polariser analyser set at a precise 0, 90, 45 and -45 degree orientation (Fig. 3) sandwiched between a garment of photo-diodes and a microlens array.

This setup not only allows instant pixel position decoded discrimination of L-CPL and R-CPL photons emitted from a preselected spectral range, facilitated by precise fast axis orientation of the QWP to a pre-aligned polarisation angle (0 degrees), but is also capable of simultaneous light detection regardless of circular polarisation state. In essence, all components required to calculate an EDCC image are encoded in one captured full field of view (FOV) frame. This is unprecedented and could potentially constitute a paradigm shift in the

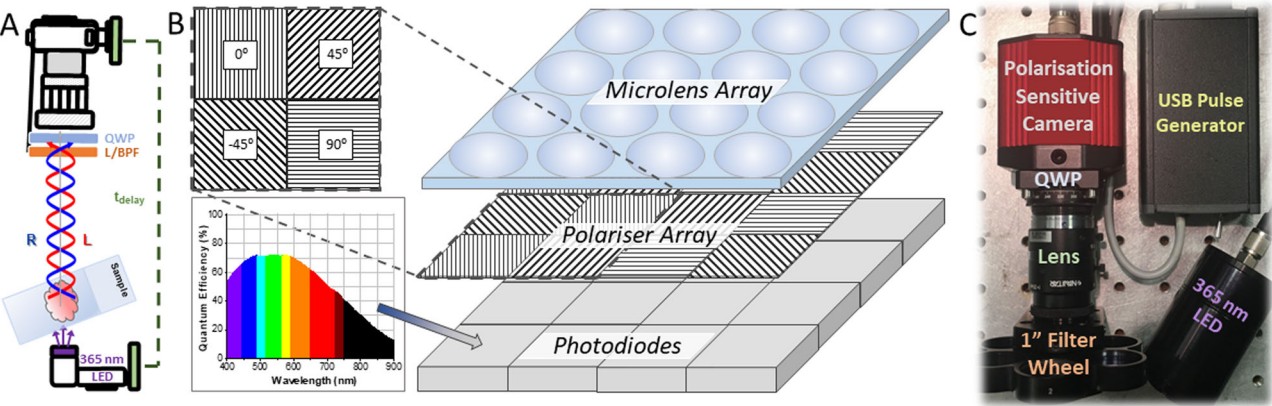

**Fig. 3 | All solid-state camera for EDCC TR-CPLP. A** Schematic diagram of the camera system and its time-resolved image capturing methodology. **B** Detailed schematics of the pixel layout and polarisation selector wire-grid alignment of the Kiralux® polarisation sensitive CMOS USB camera, displaying its wavelength-dependent quantum efficiency (ThorLabs). **C** Picture of the apparatus with its synchronised 365 nm LED (5.2 V, 500 mW) flash illuminator. The handheld USB powered apparatus comprises of a linear polarisation sensitive externally triggered camera (Kiralux®), a precisely aligned broad wavelength quarter waveplate (QWP, $\lambda_{ex} = 400$–$800$ nm), a machine vision objective lens ($f = 25$ mm/F1.4) and a selectable filter wheel containing long-pass or narrow band-pass filters (L/BPF, $d = 1$”) to achieve chromatic discrimination.

field of CPL research and its widespread establishment in the field of life and material sciences, such as incorporation of bright CPB luminophores in security ink applications. Total emission mapping of the frame is captured extracting pixel intensity values attributed to the sum of -45 and +45 degree wire grid alignment, with individual L-CPL and R-CPL captured using pixels with 0 and 90 degree polarisation orientation respectively. In practice this is facilitated by recording an image with the camera in quad-view mode, where each of the four wire grid alignment attributed individual 8-bit image is presented as an array 2 x 2 images in one frame. A simple post image acquisition protocol using a custom macro written in ImageJ (v.1.49r) crops the individual images and subsequently calculate the total emission and both (L-R and R-L) EDCC final images. The resultant three images are then saved as separate 8-bit greyscale images. In order to fulfil the last remaining design criterion of time-resolved detection, the CPLP camera system has been synchronised to a pulsed 365 nm LED. This is facilitated by the cameras native PDX/bulb exposure trigger mode linked to the camera's global shutter operation circuit. The time gating sequence has been established and miniaturised in a USB (5.2 V) or battery powered custom built signal generator[43,44]. Due to the documented rise and fall time of the LED, 11 and 6 µs respectively, the time delay ($t_d$) has been set to 20 µs (incorporating the camera's inherent 13.72 µs post-exposure ($t_{acq}$) integration period) to allow gating of any short-lived organic fluorescence out from long-loved Ln(III) emission in the spectral window of interest. To provide maximum customisation and to harness the variable triggered total accumulated acquisition time of 0.027 ms to 14 s, the signal generator has been designed with variable illumination (up to 1 s) sequences operating at 0.1, 1, and 10 Hz overall frequency.

This integrated USB powered solid-state CPLP camera system has been extensively tested and validated using our library of legacy samples (Eu:L2 - 5) previously used for validating and testing both our CPL-LSCM and CPL-epifluorescence microscope[18,19], and newly constructed complex proof of concept (POC) CSI test targets using the newly synthesised (Eu:L1) and commercial organic fluorophores anthracene, fluorescein and rhodamine B (Supplementary Fig. 4). The underlying applications of all-in-one image acquisition and EDCC decoding of the CPLP system will be a paradigm shift in chiral detection technologies and will allow CPL active Ln(III) complexes to achieve their full potential as intelligent security inks and will also open new research avenues for the multi-disciplinary research community in both life and material sciences.

**Enantioselective CPL photography (CPLP) of legacy samples and emitter Eu:L1.** To demonstrate CPLP on well controlled targets we first revisited the test slides used to validate our recently published CPL-LSCM instrument (Supplementary Fig. 10)[19]. These samples comprise of paper blotted test targets of Eu:L3 where both rough and smooth sides of the paper have been studied alongside the PVP matrix embedded test target of Eu:L3 (Supplementary Figs. 9–13). Sample preparation techniques are described in more detail in our publication describing the CPL-LSCM[19]. All studies revealed that our newly constructed CPLP allowed facile discrimination of short-lived organic fluorescence from long-lived lanthanide emission during time-resolved detection with the simultaneous capability to differentiate between Δ and Λ enantiomers of the same high CPB Eu(III) complex. This ability to encode such a high amount of photophysical information in only one captured image could render both CPLP, bright enantiopure lanthanide complexes, and high CPB organic emitters to be employed as invisible security features. One important, previously documented observation needs to be emphasised. As it has been found during our CPL-LSCM development, thin films deposited on glass surfaces can be problematic to study due to the randomisation of CPL light upon reflection from jagged or smooth reflective surfaces. This is because unlike linear polarised light, which reflects by preserving its polarisation, cork-screw light with unidirectional helicity is reversed upon reflection. We have also found with CPLP that when studying thin films, without finite optical sectioning (such as CPL-LSCM), overexposure to the excitation light source causes solarisation-induced randomisation and perpendicular light guidance of the emitted CPL (Supplementary Fig. 14) around the lateral plane edges of the sample that ultimately results in loss of CPL helicity dominance, effectively randomising CPL.

In order to emphasise the robustness of both CPLP and its detection methodology and to demonstrate the longevity of high CPB enantiopure test targets that can potentially be employed as part of CSIs. We have successfully recorded and replicated our findings using the same sample of paper blotted enantiopure Λ and Δ Eu:L4 that we have used to demonstrate EDCC for the first time in 2016 (Fig. 4)[38]. Using the calculated EDCC images: L-R shows clear CPL based emission only for Δ -Eu:L4 whilst R-L shows clear CPL based emission only for Λ -Eu:L4 demonstrating the ability of our CPLP system to be used as a facile rapid methodology for enantioselective CPL readout.

Time-resolved CPLP has also been used with thin film simulations of Chameleon Security Inks (CSI) containing enantiopure CPL emitters. We have recorded CPLP images of Λ- and Δ- Eu:L1 embedded into a

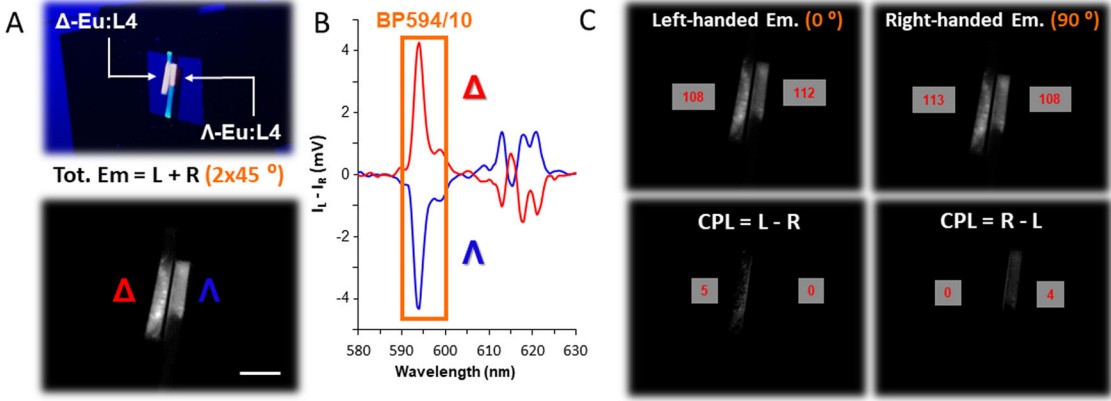

**Fig. 4 | Solid state time-resolved EDCC photography of a CPL active Eu(III) complex on paper substrate. A** (top) Conventional photo and (bottom) time-resolved CPLP of Λ and Δ-Eu:L4 in embedded into optical brightener free paper (c = 5 × 10⁻⁶ M) using 365 nm UV illumination from the smooth side of the paper. Scale bar = 0.5 cm. **B** CPL emission spectra of Δ- (red) and Λ- (blue) enantiomers of Eu:L4 in MeOH ($\lambda_{exc}$ = 365 nm) highlighting the spectral window selected for

photography using an BP594/10 (OD4.0) filter. **C** Time-resolved ($t_d$ = 20 µs) Images extracted from the quad polarisation view camera highlighting the recorded right- and left-handed emission with respect to the built-in polariser orientation to the fixed QWP fast axis. Numbers in red are avg. 8-bit pixel intensity values for each image region, $t_{acq}$ = 300 ms, 10 avg. image, 100 total image accumulation.

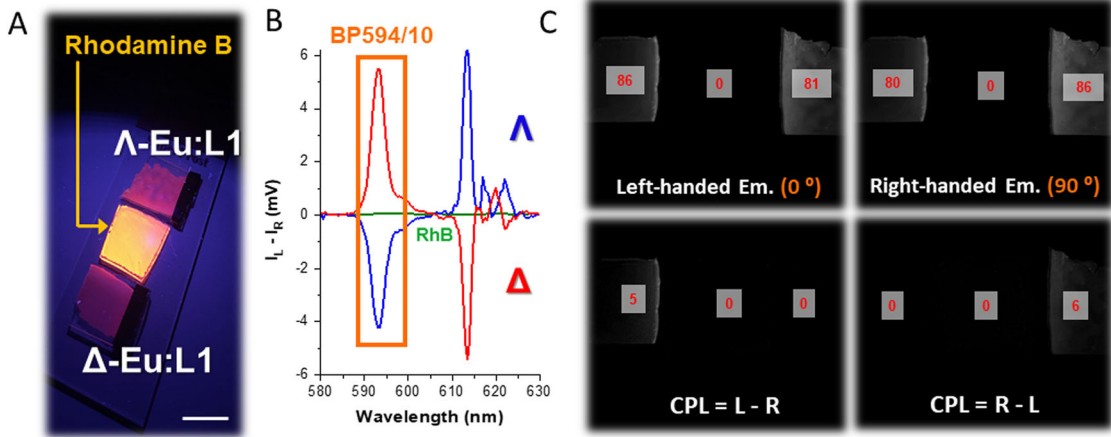

**Fig. 5 | Solid state time-resolved EDCC photography of an organic emitter and a CPL active Eu(III) complex. A** Conventional photo of rhodamine B (RhB) and Λ- and Δ -Eu:L1 in embedded into a PMMA matrix (C = 3 × 10⁻⁶ M) using 365 nm UV illumination. Scale bar = 1 cm. **B** CPL emission spectra of (green) rhodamine B, Δ- (red) and Λ- (blue) enantiomers of Eu:L1 in PMMA ($\lambda_{ex}$ = 365 nm) highlighting the spectral window selected for photography using an BP594/10 (OD4.0) filter.

**C** Time-resolved ($t_d$ = 20 µs) Images extracted from the quad polarisation view camera highlighting the recorded total emission, right- and left-handed emission with respect to the built-in polariser orientation to the fixed QWP fast axis. Numbers in red are avg. 8-bit pixel intensity values for each image region, $t_{acq.}$ = 400 ms, 10 avg. image, 100 total image accumulation.

PMMA matrix with an analogous thin film prepared using the short lived, organic, CPL inactive fluorophore rhodamine B (Supplementary Fig. 4A). Using this POC-CSI test target with just one captured image with the CPLP system revealed that under time-resolved conditions no organic emission is detected (Fig. 5 and Supplementary Figs. 15–19). Furthermore, whilst using the calculated EDCC images L-R shows clear CPL emission arising from the Δ enantiomer of Eu:L1 only, whilst R-L shows CPL emission only from the Λ enantiomer of Eu:L1, demonstrating the capability of CPLP.

The results show that steady state and time-resolved EDCC has been recorded with all samples studied. Each sample measured a 1 cm x 1 cm area with clear enantioselective differentiation of Δ- and Λ-enantiomers based on the left- or right-handed emitted CPL dominance of the selected spectral window of interest (BP594/10, 589–599 nm or BP610/10, 605–615 nm). With the employment of the optional time-resolved detection, short lived organic fluorescence has also been completely eliminated. This effect can be used to give rise to

contrast based optical separation or 'chameleon-like colour separation' of short- and long-lived emissive ink components. It is important to emphasise that similar TR-CPLP EDCC results can be easily achieved using green emitters, for example the Λ- and Δ-enantiomers of a Tb(III) complex alongside the organic dye fluorescein (Supplementary Fig. 4C, D) with the detection window spectrally tuned to its magnetic dipole allowed high CPB $\Delta J$ = 5 (BP 546/10 nm, 541–551 nm) band (Supplementary Figs. 21, 22)[45].

To comprehensively demonstrate the validity of our claim that the all solid state one-shot CPLP camera system developed herein is capable of five layers of security readout comprising of multi-coloured, multi-spectral, opposing-helicity, combined with high spatial and temporal resolution we have constructed a proof-of-concept (POC) invisible hidden in plain sight CSI security tag. This CSI tag (Fig. 6 and Supplementary Figs. 4, 23) was constructed using three different coloured - anthracene (blue), fluorescein (green) and rhodamine B (red) - commercially available short-lived organic fluorophores and

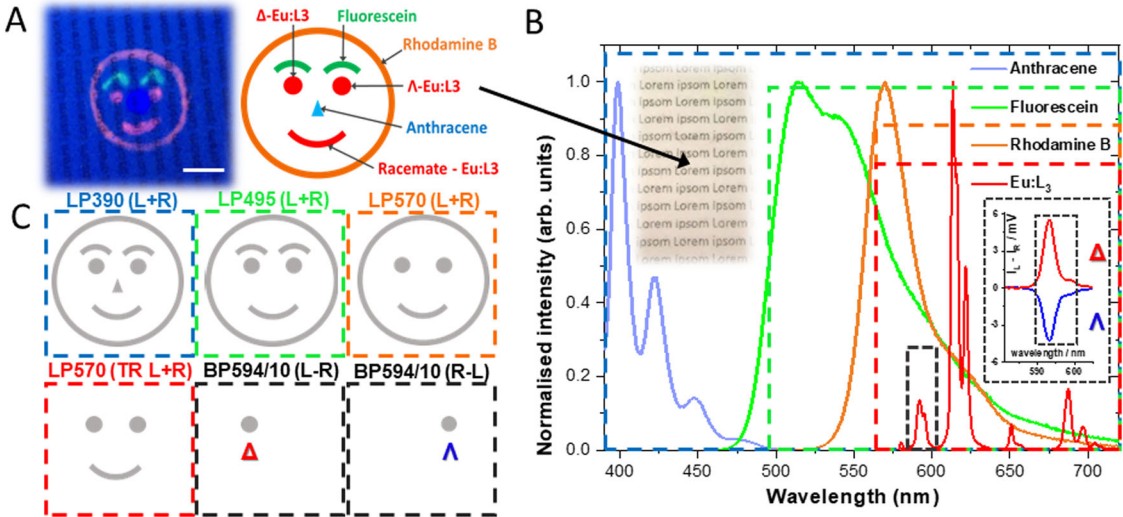

**Fig. 6 | Design and composition of the POC CSI security tag. A** Schematics of the fluorophore composition and UV (365 nm, standard document reader) illuminated actual image of the concept invisible rudimentary Chameleon (CSI) security tag, scale bar 1 cm. **B** Individual emission spectra of the CSI's luminophores, highlighting the colour coded emission filters used for revealing five tiers of embedded security features. **C** Detection methodology and proposed CPLP imaging sequence of the POC CSI security tag. One image encodes six different images that can be individually revealed. Chromatic separation can be facilitated using a combination of appropriate bandpass filters and time-resolved (TR) detection as depicted using CPLP, whilst opposing helicity information can be read out using individual CPL channel detection.

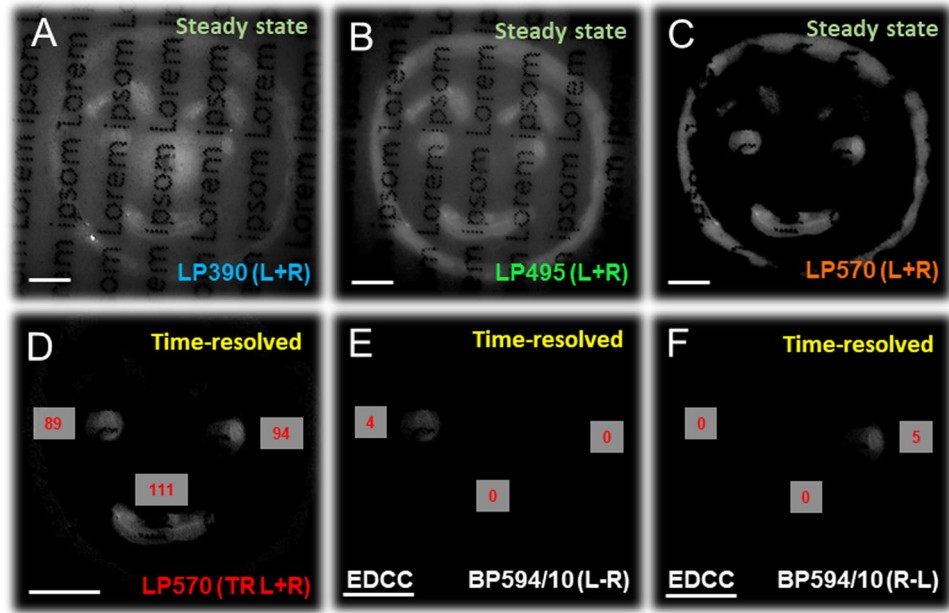

**Fig. 7 | Decoding the RGB EDCC layering of the laminated POC CSI security tag.** Images of the POC CSI security tag ($\lambda_{exc}$ = 365 nm) using various colour and time resolution filtered CPLP. **A–C** Total emission under steady state illumination using a (**A**) LP390 nm (**B**) LP495 nm and (**C**) LP570 nm filter. **D** Time-resolved image ($t_d$ = 20 μs) using a LP570 nm. **E** L-R EDCC image using a BP594/10 nm filter demonstrating clear CPL based emission for Δ-Eu:L1 whilst (**F**) R-L shows clear CPL based emission for Λ-Eu:L1. Numbers in red are avg. 8-bit pixel intensity values for each image region, $t_{acq}$ = 70–700 ms, 10 avg. image, 1–100 total image accumulation. Scale bar = 2 mm.

enantiopure and racemic (50-50) mixture of Λ- and Δ-Eu:L1. The individual components (C = 2.8 × 10$^{-6}$ M) of this CSI dye mixture were embedded onto an optical brightener free paper substrate under a standard UV document reader (365 nm lamp), with each layer homogeneously overdrawn multiple times to ensure that all tag feature appears to be of equivalent brightness to the naked eye. To add an extra layer of complexity, the paper has been pre-printed with black text prior to security tag construction using a commercial laser printer and the final POC CSI tag have been laminated using a standard PET/ EVA (polyethylene terephthalate/ethylene-vinyl acetate) laminating sheet and a laminator at 150 °C (1 cm/s) and left in the dark for three days prior to CPLP imaging.

To reveal the POC CSI tag's invisible five tiers of security features we have recorded a sequence of five one-shot CPLP images. We recorded three steady state (i.e., non time-gated) images (Fig. 7A–C) using chromatic RGB (red-green-blue) separation, facilitated by appropriate long pass filters - LP390 nm for RGB, LP495 nm for RG and LP570 nm for R. We then employed a time-resolved image collection mode, using the

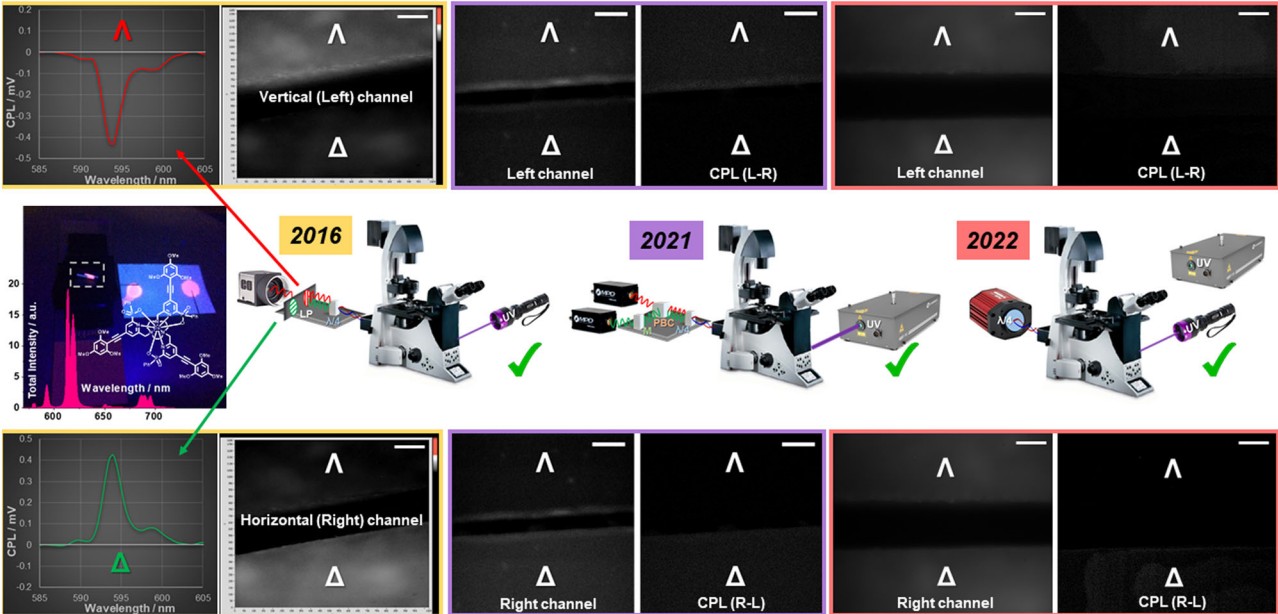

**Fig. 8 | Chronological evolution and schematic summary of CPL microscopy and the associated chiroptical separator layouts.** (2016) The first epifluorescence microscope capable of chiroptical separation of left- and right-handed CPL using a quarter wave plate (QWP) and orthogonally pre-aligned linear polarisers (LP). Identical results were recorded using fixed LP and orthogonally aligned QWP positions. (2021) The world's first laser scanning confocal microscope capable of simultaneous dual channel EDCC imaging using a QWP, achromatic broad wavelength (400–800 nm) beam splitter cube and two LP assisted (horizontal and vertical LP) single point HyD avalanche photodiode detectors. (2022) EDCC microscopy using the solid-state single chip CPLP camera capable of recording one frame full FOV EDCC imaging on an epifluorescence microscope setup. The individual microscopy images shown are achieved using the same enantiopure Eu:L4 paper-based microscopy slide (Fig. 4 and Supplementary Fig. 10) using a BP594/10 filter. Scale bars = 30 μm. (Microscope schematics are adapted from ref. [18], CC BY 3.0, with permission from the Royal Society of Chemistry.).

same red LP570 filter, to gate out the short-lived red (rhodamine B) feature of the tag, revealing embedded additional 'secondary red' features due to long-lived Eu:L3 emission (Fig. 7D).

Finally, a fifth time-resolved image has been recorded (Fig. 7E, F and Supplementary Fig. 24) to reveal the left- or right-handed emitted CPL dominance of the employed enantiopure Eu(III)-complex features using a BP594/10 nm filter. The calculated EDCC L-R image shows clear CPL emission arising exclusively from the Δ enantiomer, whilst the R-L image shows CPL emission only from the Λ enantiomer of Eu:L1.

Using this rudimentary POC CSI invisible tag we have demonstrated the capability and simplicity of CPLP. We must emphasise two key points. Firstly, the imaging sequence and chromatic separation element can be further simplified by omitting the red channel (LP570 nm filter) element and only recording the steady state or time resolved red CPL spectral region of interest (BP594/10 nm) directly (Supplementary Fig. 24). EDCC images can still be calculated with high accuracy as the differential chiral contrast of a bright CPB luminophore is independent of the applied time domain (steady state *vs* time-resolved) mode of collection of CPLP (demonstrated in Fig. 5 and Supplementary Figs. 19, 24).

Secondly, the POC CSI tag has been constructed using stencils and calligraphy pens (Supplementary Fig. 23). However, this could have been made using inkjet printing adaptations for large scale commercial purposes. Furthermore, an extra dye and subsequent additional image feature could be introduced in the form of a second time-resolved green ink containing Tb:L6 (Supplementary Figs. 21, 22) using an additional BP546/10 nm filter. To ensure simplicity when presenting monochromatic EDCC POC images (Fig. 7 and Supplementary Fig. 24), we have deliberately omitted this extra layer.

**Adaptation of CPLP to microscopy.** Owing to its compact and robust nature, the CPLP camera system described herein renders itself to be

an excellent candidate as a simple add-on detector for the next generation of CPL-microscopes allowing EDCC. We have extensively tested the camera system with both our epifluorescence[18] and CPL-LSCM (Fig. 8)[19]. EDCC analysis has been successfully achieved using our enantiopure test target samples with both types of microscopy techniques mentioned above, coupled to CPLP. CPLP-epifluorescence microscopy could be a simple and cost-effective full field of view (FOV) image acquisition alternative to CPL-LSCM with close to diffraction limited lateral resolution that is coupled to limited optical sectioning capability (axial resolution) due to the absence of confocality. Nevertheless, it will allow the broad multidisciplinary research community to achieve EDCC images in both life and material sciences. For further discussions regarding CPLP as a detector for biological imaging applications using LSCM refer to the dedicated section in the Supplementary Information (page 24).

## Discussion

To conclude, we have presented three new major milestones in the intertwined disciplines of CPL detection, chiral luminescence imaging, and enhanced security applications. First and foremost, we have developed a handheld non-moving part CPLP system capable of rapid and simultaneous differentiation of left- and right-handed emitted CPL and allows one-shot full field of view enantioselective differential chiral contrast imaging and photography. This is accompanied with the inherent advancement of differentiation between short-lived organic and long-lived lanthanide emission via time-resolved detection. Secondly, we have created a simple protocol to generate mixed chemical entities to act as rudimentary proof-of-concept chameleon security inks, encompassing five tiers of security, where both time-resolved colour separation and pattern recognition can be facilitated with the added possibility of further patterning using EDCC. These test targets consist of enantiopure high CPB europium complexes specifically designed to have exclusively positive or negative sign CPL within an

emission manifold embedded into PMMA matrix. Deposition of thick films was achieved by calligraphy or drop casting and spin coating. To evidence the adaptability of CPLP we have constructed and studied a rudimentary POC CSI security tag. Using this invisible to the unaided eye tag we have demonstrated the capability and simplicity of CPLP and its of 5 layers of chromo-spatio-temporal-spectral-chiral discrimination. Importantly, these simple test target generation protocols can serve to compare and validate future CPLP devices capable of EDCC that may subsequently be developed by the multidisciplinary imaging community. Finally, we have demonstrated that by harnessing the simplistic yet radical design of our solid-state CPLP, EDCC can be facilitated on the microscale by creating a cost-effective yet functional lower optical resolution alternative to our recently developed CPL-LSCM[19]. This microscope has also been validated using our new and legacy EDCC test targets underpinning the robustness and resilience of Ln(III) CPL emitters to be used as security ink components.

We anticipate that these developments will ignite research avenues especially in the field of lanthanide chemistry and luminescent security ink development for enhanced security applications. CPLP will play a vital role in the new era of layered information photography enabling the development of physically unclonable stochastically micro-patterned CPL- active chameleon security inks[1,27,46] and will promote EDCC to be adopted and utilised by a wider research community. Ultimately the work presented here could promote sufficiently high CPB luminescent chiral molecules to be used in many aspects of materials and life science. The development of a such compact multifunctional camera system and image differentiation methodology can also take centre stage and dictate the development and applications of chiral molecular emitters.

## Methods

### Enantioselective Differential Chiral Contrast (EDCC) photography

The CPLP handheld instrument for EDCC comprises of an externally triggered Kiralux® polarisation 5.0 Megapixel CMOS USB camera (CS505MUP1, Thorlabs). It is synchronised to a 365 nm LED (Nichia, 5.2 V, 500 mW) flash illuminator driven by an internal custom built signal generator module operating at master frequencies of 0.1, 1 or 10 Hz. This device has been constructed to be able to provide both a variable illumination (1 ms–1 s) and image acquisition (0.027 ms–14 s) sequence with a constant 20 $\mu$s time delay between the two pulses to allow discrimination of short-lived organic (ns-$\mu$s) and long-lived lanthanide pseudo phosphorescence (ms). The chiroptical separator of the apparatus comprises of the built in pixel decoded wire grid polariser array complemented by a precisely aligned broad wavelengths ($\lambda = 400$–800 nm) quarter waveplate (QWP, Thorlabs AQWP05M-600), a machine vision (Navitar $f = 25$ mm/F1.4) objective lens (interchangeable to a LWD $f = 50$ mm/F2.8 lens or 0.63x video relay lens (Edmund Optics) for epifluorescence microscopy) and a selectable filter wheel containing high precision narrow range band pass filters (such as Edmund optics BP546/10, BP589/10, BP594/10, and BP610/10) to achieve chromatic discrimination.

Camera control and image acquisition were performed with ThorLab's commercial camera software ThorCam™ or an adapted custom LabView code to facilitate time-resolved detection and controlled external camera triggering[43]. The camera was operated in quad view (Supplementary Fig. 20) where the 16-bit overlayed total image has been split up into four individual 8-bit images decoding each wire grid polariser state orientation (0, 90, 45, and -45 degree) captured image as a 2 x 2 array in one captured frame. These images were generated by area defined crop and paste individual image generation without pixel position reassignment. Such as the previously reported EDCC using our recently developed CPL-LSCM, this one step resolution preserving (lossless compression) image processing is achieved using a custom written script (macro) in ImageJ (v1.49)[47].

Enantioselective differential chiral contrast (EDCC) imaging was also built into our custom written macro 2.0 or can be post processed using ImageJ's built-in image calculator add-on software by subtracting one CPL channel from the other, and vice versa. The convention used herein is: left-handed enantioselective contrast = left CPL - right CPL; right-handed enantioselective contrast = right CPL - left CPL. Images were recorded with the camera's native 2448 x 2048 pixel field of view (FOV) without image cropping with total accumulated integration time varied from 20 ms to 1 s with 5–32 frame averaged sequences. The total integration time has been determined case by case by careful monitoring of maximum 8-bit pixel values to eliminate pixel saturation and achieve maximum overall image brightness for EDCC calculation.

The all-important 8-bit average pixel chiroptical contrast value calculations were facilitated by selecting and averaging five different non-overlapping, equal size and shape, arbitrary area portions of the sample with respect to each enantiomer and dark background. Due to the 2448 x 2024 pixel size of each recorded image total field of view (FOV), this arbitrary area has been kept at a constant size of 200 x 200 pixels region of interest (ROI). The average maximum 8-bit greyscale pixel intensity values were determined using ImageJ Analyse-Measure macro with mean ROI intensity value mode that employs a maximum average value ROI histogram methodology that is based on a standard Gaussian distribution profiling of the average intensity values. Due to the employed methodology and the averaging nature of image acquisition and ROI calculation, the limit of detection (error associated with) 8-bit greyscale contrast value is below 1% (< 3 average greyscale value/pixel on a 0–255 pixel intensity scale). This is determined using the total europium emission image and the selected arbitrary ROI area selection is then kept identical throughout the imaging sequence resulting in high-precision chiroptical contrast calculations.

### Statistics and Reproducibility

Where instruments incorporating a scanning monochromator have been used (absorption, emission, and excitation spectra) each sample has been recorded and averaged as triplicate measurements. Spectra, where CCD detectors have been employed, such as two-photon cross-section determination, multiphoton, and CPL spectroscopy, have been measured as an average of one thousand spectra on triplicate samples.

Photography images presented herein are representative images of the experiments discussed. Each experiment has been repeated in a minimum of five separate imaging sequences.

Imaging parameters presented, such as brightness and chiroptical contrast have been calculated on each individual imaging sequence according to the protocol detailed in the Supplementary Information (SI) section.

### Reporting summary

Further information on research design is available in the Nature Portfolio Reporting Summary linked to this article.

## Data availability

All data generated and analysed during this study including spectra, high-resolution microscopy and photography images and drawings are available from the corresponding author upon request.

## Code availability

Custom codes written and developed and used during this study are available from the corresponding author upon request.

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

## Acknowledgements

R.P. acknowledges support from the Royal Society University Research Fellowship URF\R\191002, H2020-MSCA-ITN-859752 HEL4CHIROLED and BBSRC BB/S017615/1 and BB/X001172/1, RP thanks Prof. David Parker and Prof. Andrew Beeby for the stimulating discussions regarding CPL active compounds and CPL spectroscopy; Dr. Andrew Frawley for the synthesis of Eu:L2-5 complexes used herein, Dr. Jack Fradgley for the provision of some precursors used for the synthesis of Eu:L1, Paolo Mastroeni for the resynthesis of Eu:L3 and Thomas Bradford for assisting with sample preparation. Kelvin Appleby for the construction of the USB-powered signal generator box. We also express our gratitude to Dr. Lewis E. MacKenzie for his work on the prototype rapid dual-channel CPL spectrometer.

## Author contributions

D.D.R.: Synthesised and provided lanthanide complexes for analysis, helped developing the final camera layout, and drafted/edited the manuscript. P.S.: Prepared and provided heritage lanthanide microscopy samples for analysis, performed multiphoton spectroscopy, and edited the manuscript. D.J.B.: Helped to prepared POC CSI test target, helped developing the final camera layout and during imaging experiments, and edited the manuscript. R.P.: Secured project funding, developed and constructed the CPL camera concept and associated detection methodology, performed all imaging experiments, and drafted/edited the manuscript.

## Competing interests

R.P. is inventor on filed patent WO2016174395A1: Light detecting apparatus for simultaneously detecting left-and right-handed circularly polarised light. There are no other competing interests.
