## [Peer Review File · Nature Communications]

Rapid handheld time-resolved circularly polarised
luminescence photography camera for life and material
sciencesReviewer #1 (Remarks to the Author):

In this work, CPL spectroscopy of the luminescence of a rare earth complex, which has been proposed as a security ink, is performed using a high-speed scanning CPL microspectroscopic device developed by the author,. The argument is that the measurement can be done in a short time by improving the equipment, and that the difference in circularly polarized light can be measured quickly.. Because these cannot be distinguished by the naked eye, security printing is available using this systems and emitting ink. The CPL spectroscopy proposed by them is very interesting, but lacks novelty because it has already been discussed (ref.13). In addition, it is claimed that it can be used for security printing, but it does not seem to have reached the level to be published in NatureChem, which requires articles that strongly evoke a wide range of natural science fields.

Reviewer #2 (Remarks to the Author):

In this paper, the authors reported the development of third-generation circularly polarized luminescence (CPL) photography camera system. By adopting polarisation sensitive scientific cameras (a Kiralux® CS505MUP1, ThorLabs) with 4 pixels masked with a wire grid polariser set at four orientations, the camera can simultaneously record L-CPL and R-CPL of photons and total photons, and rapidly obtain enantioselective differential chiral contrast images of enantiomeric Eu-complexes by calculating L-R or R-L. Indeed, compared with the last two generation products, it presents the huge advance. As the authors' proposed that if third-generation camera is used in "Chameleon Security Inks (CSI)", it will offer 5 layers of security once the blend of long-/short-lived emitters, because it can offer multi-spectral, opposing-helicity, combined with high spatial and temporal resolution. It is my opinion that the manuscript well-deserves publication on a top Nature journal in view of its practical application in life and material sciences. I suggest that the challenge for this camera is the difficult application in determination of luminophores with little glum value, except for the lanthanide complexes.

some issues:

1. The calculations for CPB in Figure S1 are incorrect, (1) as using eq.2 "gem" dose not divide by 2; (2) it cannot use the total luminescent quantum yields of 5D0 transitions to displace the 5D0 to 7F1 transition.
2. Authors should introduce why to design the new complex Eu:L1 in manuscript.

Reviewer #3 (Remarks to the Author):

This short, but ultra-technique communication entitled "Rapid handheld time-resolved circularly polarised luminescence (CPL) photography camera for life and material sciences" has been submitted to Nat. Comm. This reviewer carefully read this contribution and congratulate the authors for producing a valuable issue with a top-quality scientific level. However, this modest referee thinks that the degree of novelty and originality, compared with the previous publications of the same team: Chem. Commun., 2016, 52, 13349, Nat. Commun., 2020, 11, 1676 and Nat. Commun., 2022, 13, 553, does not reach the threshold to be published in Nat. Commun. A real-life example such as implementing the chiral Eu chromophore as a security ink, and using their technology to detect the CPL would be interesting and novel.

In any case, few point should be address for any further submission:

1. In table SI 1 there are few mistakes concerning the calculated CPB. If using the formula: $CPB = \xi_{abs} \phi_{em}(gem/2)$, the calculated CPB values obtained by this reviewer for Eu:L1, Eu:L2, Eu:L3, Eu:L4, Eu:L5 and Eu:L6 are not in agreement with the reported by the authors. The values obtained are half of those reported. Please check the numbers.
2. In the SI, authors have mentioned that they used two standards for the quantum yields measurements. Please add the chemical name/formula of those standards.
3. The authors mention "Chiral HPLC analysis" in the main text and SI but they did not show any chromatogram neither in the main text nor in the SI. Please add the chromatograms of the Eu:L1.

Authors Response to Reviewers' Comments

Collective response summary to the Reviewers:

We greatly appreciate the reviewers' constructive comments and have attempted to address them in detail in the revised manuscript. As underlined in the manuscript one of the forecast mainstream applications of CPL could be in invisible security inks. However, the lack of a small, simplistic, yet robust and versatile 'snapshot' camera system prevented this until now. In order to support this further have taken on board the reviewers' comments and have included two new figures in the main text (Figures 6 and 7) and six new figures in the SI (Figure SI 1, 4, 8, 9, 23 and 24) to clearly demonstrate and emphasize the capability and ground-breaking nature of our camera to distinguish to facilitate ad hoc time-resolved EDCC based one-shot CPL photography via chromo-spatio-temporal-spectral-chiral discrimination *via* a proposed real life 'rudimentary' POC CSI security ink design.

New Figures Included:

Figure 6. Design and composition of the proof-of-concept CSI security tag. (A) Schematics of the fluorophore composition and UV (365 nm, standard document reader) illuminated actual image of the concept invisible rudimentary Chameleon (CSI) security tag, scale bar 1 cm. (B) Individual emission spectra of the CSI's luminophores, highlighting the colour coded emission filters used for revealing five tiers of embedded security features. (C) Detection methodology and proposed CPLP imaging sequence of the POC CSI security tag. One image encodes six different images that can be individually revealed. Chromatic separation can be facilitated using a combination of appropriate bandpass filters and time-resolved (TR) detection as depicted using CPLP, whilst opposing helicity information can be read out using individual CPL channel detection.

Figure SI 23. Construction and physical composition of the POC CSI security tag (A) Conventional photo of the final laminated (150 °C) optical brightener-free paper with laser printed text containing the invisible to the naked eye POC CSI security tag. (B) Hand-drawn POC CSI security tags using a calligraphy pen and individual UV (365 nm, standard document reader, LP390 nm emission cut off filter) illuminated solutions of the dyes used (from left to right) Rhodamine B in EtOH, Anthracene in EtOH, Fluorescein in MeOH, Racemate (1:1) Eu:L1, and individual enantiopure Λ -Eu:L1 and Δ -Eu:L1 in EtOAc as a carrier solvent, scale bar 1 cm. (C-E) Zoom-in image of the selected POC CSI tag for CPL photography imaging under 365 nm UV illumination (laminated, standard UV document reader, iPhone 6S camera) using (C) LP390 nm, (D) LP495 nm and (E) LP570 nm filters.

Figure 7. Decoding the RGB EDCC layering of the laminated POC CSI security tag. Images of the POC CSI security tag ($\lambda_{exc} = 365$ nm) using various colour and time resolution filtered CPLP. (A-C) Total emission under steady state illumination using a (A) LP390 nm (B) LP495 nm and (C) LP570 nm filter. (D) Time-resolved image ($t_d = 20$ μ s) using a LP570 nm. (E) L-R EDCC image using a BP594/10 nm filter demonstrating clear CPL based emission for Δ -Eu:L1 whilst (F) R-L shows clear CPL based emission for Λ -Eu:L1. Numbers in red are avg. 8 bit pixel intensity values for each image region, $t_{acq} = 70$ -700 ms, 10 avg. image. Scale bar = 2 mm.

Figure SI 24. Steady state and time-resolved EDCC photography images of the laminated POC CSI security tag. Images extracted ($\lambda_{exc} = 365 \text{ nm}$, $\lambda_{em} = 589\text{-}599 \text{ nm}$ using a BP594/10 filter) from the quad polarisation view camera highlighting the recorded total emission, right- and left-handed emission with respect to the built-in wire-grid polarised orientation to the fixed QWP fast axis. Total emission under (A) steady state and (D) time-resolved ($t_d = 20 \mu\text{s}$) conditions. (B) Left- and (C) right-handed emission under steady state condition compared to (E) left- and (F) right-handed emission under time-resolved ($t_d = 20 \mu\text{s}$) image collection mode. The calculated EDCC image (G) L-R shows clear CPL based emission only for $\Delta\text{-Eu:L1}$ whilst (H) R-L shows clear CPL based emission only for $\Lambda\text{-Eu:L1}$ and demonstrates that EDCC is independent of the applied time domain (steady state vs. time-resolved) mode of collection of CPLP. Numbers in red are avg. 8 bit pixel intensity values for each image region, $t_{acq} = 700 \text{ ms}$, 10 avg. image. Scale bar = 2 mm.

All new additions are highlighted in yellow in both the revised manuscript and SI.

Point by point response to the individual reviewers' comments:

Reviewer #1 (Remarks to the Author):

(Comment 1) In this work, CPL spectroscopy of the luminescence of a rare earth complex, which has been proposed as a security ink, is performed using a high-speed scanning CPL microspectroscopic device developed by the author. The argument is that the measurement can be done in a short time by improving the equipment, and that the difference in circularly polarized light can be measured quickly. Because these cannot be distinguished by the naked eye, security printing is available using this systems and emitting ink. The CPL spectroscopy proposed by them is very interesting, but lacks novelty because it has already been discussed (ref.13)

Response to Reviewer 1 Comment 1

We thank the reviewer for the comment, however, we respectfully disagree and feel that simply comparing the novelty of our camera system (capable of 5 layers of chromo-spatio-temporal-spectral-chiral discrimination) to 3D-glasses using pre-aligned QWP-LP garments greatly dismisses our hard work and recent achievements.

For context, as it has been detailed in the manuscript in the last 7 years, our research has revolutionised the field of CPL detection by significantly changing the 50-year-old (Gafni, Review of

Scientific Instruments 43, 409 (1972)) sluggish, large footprint instrument design that is a major limiting factor in CPL research. Our publication sequence in the last 7 years has cemented our reputation as the instrumental pioneers of the field and they generated significant interest world-wide by other researchers adapting our technology and methodology. We have pushed CPL instrumental development to previously uncharted territories by not only developing and demonstrating the world's first all solid-state CPL spectrometer (2021), Epifluorescence (2016) and Laser Scanning Confocal (2022) CPL microscope, but also recorded the first ever 2 photon activated CPL spectrum. All these achievements have been triggered by continuously adapting our detection technology, step-by-step improving and simplifying the components used. Despite these components being available to everyone, we were the only one to have had the novel idea to combine them for this purpose.

We sincerely hope that after our scientifically underpinned reasoning (*vide infra*) the reviewer now appreciates that the evolution of our spectroscopy followed by microscopy and now time resolved multicolour photography application of the QWP-LP garment is nontrivial and indeed extremely novel.

To support this, we would like to present 3 arguments:

1. Novelty often involves constructing something using readily available components (known polarisation altering devices) in an unprecedented sequence and manner by carefully tailoring one to another. The camera design detailed in our manuscript appears to be uncomplicated, but we believe this simplicity is one of its main strengths and fulfils clear and unmet need. To our knowledge no one has ever constructed an all-solid state non-moving part small footprint CCD or CMOS based handheld CPL camera or indeed has used a polarisation sensitive camera for CPL research. We are the first research group in the field who managed to construct such a camera, using off the shelf components, in the hope to finally allow CPL and CPL active dyes to fulfil their full potential within the wider research community and even for large scale industrial applications, such as security inks within bank notes.
2. We would like to address the 3D glasses and their working principle. Conventional non colour-coded 3D glasses use QWP-LP garments to aid the human eye in converting a 2D image into a pseudo-3D image. This technology was devised having two simultaneous detectors in mind. This design would not immediately work with only one detector without adaptations. Even if the lens is rotated 90 degrees, it would still require two synchronised simultaneous detectors. We need to emphasise that up to this date all commercial CPL spectrometers still employ the PEM, LIA and monochromator detection system and not the QWP-LP garment. This confirms the challenge in constructing such technically complex yet conceptually simple instrumentation. Therefore, we deliberately included ref 13 to demonstrate how long it took (14 years) to adapt the core 3D glasses technology into a handheld scientific CPL camera system. Our design overcomes the issues relating to single detector usage, one-snap opposing-helicity encoded imaging, chromatic and CPL inverting artefact suppression.

Previously we used an adaptation of the core QWP and rotating LP principal in our solid-state CPL spectrometer and microscopes to separate CPL light of opposing helicity in a fast, facile manner, but using automated rotation mounts. Our camera system's four different orientation of LP array wire grid allow the user to decode 4 separate images. For the first time ever, this non-moving part containing hardware setup provides the end user with a one snap solution to record total emission and individual left- and right-handed emission using one detector. Applying the principles of ref 13 and rotating the lens from 3D glasses in front of a camera (Such as Kitagawa's 2016 work, that followed our 2016 publication) would only allow this to be recorded in 4 separate frames and, most importantly, it will require precise pairs of orthogonally aligned 3D-lens

garments. This results in undesired complexity and in the introduction of artefacts (as we detailed these drawbacks in our 2016 publication, spurring our later developments). We need to yet again emphasise the simplistic novelty that four images (the +45 and -45 channels to provide total emission and the +90 and 0 channels for L-CPL and R-CPL respectively) are needed to eliminate any sources of error when calculating the total emission from recording only left and right CPL images.

- 3.** We would like to highlight that we have been granted a patent (WO2016174395A1, 2016) based on the core QWP-LP garment technology. This demonstrates that the invention and application knowhow is indeed novel, that it presents a key inventive step, and that there are no precedents in the prior art. This patent is explicitly mentioned in the disclaimer of all our recent CPL instrumentation papers including this one.

We therefore believe that using the concept behind cinema-style 3D glasses to invalidate the novelty of our instrumental design is an unsound argument, and we hope that the reviewers will acknowledge the arguments we have presented and reconsider their conclusion.

(Comment 2) In addition, it is claimed that it can be used for security printing, but it does not seem to have reached the level to be published in Nature Chem (*Nature Communications*), which requires articles that strongly evoke a wide range of natural science fields.

Response to Reviewer 1 Comment 2

To emphasise the novelty, elegance, and impact of our instrument and its methodology additional work has been included in the revised manuscript regarding security ink applications. In the revised manuscript we have extensively demonstrated our camera's capability to discriminate light in terms of our five exclusion criteria: multi-spectral, opposing-helicity, multi-layered, combined with high spatial and temporal resolution. Moreover, that through the inclusion of the new figures (Figures 6 and 7 and Figure SI 1, 4, 8, 9, 23 and 24) depicting a proposed real life 'rudimentary' POC CSI security ink design and its underlying five-fold security feature to be instantly read out by our CPLP camera we sincerely hope that underpinning the proposed security applications will warrant our revised manuscript's acceptance and publishing as the 3rd instalment of our CPL instrument evolution in *Nature Communications*.

We would like to emphasise that, as a testament of our CPLP camera's ease of use and rapid nature, construction of the proof-of-concept CSI test target and obtaining the images for the new figures (Figure 6 and 7, SI 23 and 24) took only one day with combined efforts and enlisting help for a now author status colleague who therefore has been removed from the acknowledgment section.

Also, we are aware that our new proof-of-concept figure (Figure 7) has been constructed using stencils and calligraphy pens – this is a playful demonstration that emphasises the simplicity of our read-out system - but in future this can be adapted for inkjet printing for large scale commercial purposes with suitable time and resources. We must emphasise that at the time of writing we do not yet have the printing instrument capability to carry out high quality inkjet printing – although we endeavour to pursue this in future. Our manuscript is purposefully constructed in a way that our proof-of-concept design can be further adapted or provide a blueprint to researchers world-wide to reproduce and advance it. We note that sourcing the required printing components and optimising printing parameters (e.g. surface tension of ink) would add very little to the fundamental POC nature of our work and delay the dissemination of our findings for the benefit of the field.

We have also demonstrated the capability of our snapshot CPL camera to a wide audience within the chemistry, materials science, and life sciences fields by not only demonstrating its application in photography of solution and solid-state test targets, and novel security-ink patterns, but by also using it as a camera for microscopical applications. Indeed, we directly compare the microscopy application to our 2016 and 2022 CPL microscopes using identical robust test targets (see Figure 8). Live cell imaging applications and adaptation of this is outside the scope of this article and the principles and feasibility of live cell CPL confocal microscopy have already been presented in our 2022 CPL-LSCM Nature Communication paper.

We hope that we managed to change the reviewer's mind and help them realise the underlying novelty of our camera and one step EDCC generation warrant publication of our work in Nature Communications.

Reviewer #2 (Remarks to the Author):

(Comment 1) In this paper, the authors reported the development of 3rd-generation CPL photography camera system. By adopting polarisation sensitive scientific cameras (a Kiralux® CS505MUP1, ThorLabs) with 4 pixels masked with a wire grid polariser set at four orientations, the camera can simultaneously record L-CPL and R-CPL of photons and total photons, and rapidly obtain enantioselective differential chiral contrast images of enantiomeric Eu-complexes by calculating L-R or R-L. Indeed, compared with the last two generation products, it presents the huge advance. As the authors' proposed that if third-generation camera is used in "Chameleon Security Inks (CSI)", it will offer 5 layers of security once the blend of long-/short-lived emitters, because it can offer multi-spectral, opposing-helicity, combined with high spatial and temporal resolution. It is my opinion that the manuscript well-deserves publication on a top Nature journal in view of its practical application in life and material sciences. I suggest that the challenge for this camera is the difficult application in determination of luminophores with little g_{lum} value, except for the lanthanide complexes.

Response to Reviewer 2 Comment 1

We thank the reviewer for the kind and encouraging comments and sharing the excitement for our work and its underlying game-changing potential in both material and life sciences.

We do agree the next challenge is to improve detection sensitivity or to employ some clever instrumental 'trickery' again, to allow its use in detecting predominantly organic luminophores with small g_{lum} values. As we detailed in the manuscript, often these luminophores, due to their low g -values, display modest circularly polarised brightness (CPB) despite having high quantum yields and high molar extinction coefficients. It is CPB that is the most important parameter for CPL imaging, and currently chiral lanthanide complexes offer CPB values 2 to 3 orders of magnitude greater than organic luminophores. However, it is also important to emphasise that the often broad single sign CPL transition of these organic luminophores allow us to employ wide (>100 nm) bandpass filters to maximise collection efficiency, and that novel strategies are being utilised to boost chiral emission from organic luminophores, so they should not be discounted as CPL-active security ink candidates.

To offer reassurance about the performance of the CPLP with organic CPL emitters, after our 2022 CPL-LSCM microscope publication we established collaborations with several research groups worldwide, mostly within the CPL spectroscopy and microscopy study of chiral organic luminophores. As part of our validation process for our novel CPLP camera we asked for their permission to try these novel chemical entities with our detection system described herein. So far two organic luminophores have proven to be suitable. However, due to the novel, unpublished nature of these compounds, our collaborators explicitly requested not to include these in the current manuscript. We are in no doubt

that once our paper is published, we will not only be in the position to include our CPLP images of these organic emitters into their respective publications but will also establish new collaborative avenues to push the detection limit of our instrument and of equal importance broaden its application and our CPL horizon.

some issues: (Comment 2) The calculations for CPB in Figure S1 are incorrect, (1) as using eq.2 “gem” dose not divide by 2; (2) it cannot use the total luminescent quantum yields of 5D0 transitions to displace the 5D0 to 7F1 transition.

Response to Reviewer 2 Comment 2

This is very embarrassing for us! We thank the reviewer (and for reviewer 3 too) for noticing this and we have of course rectified this monumentally embarrassing mistake. Additionally, we expanded the table to calculate transition (detection window) specific ‘true’ CPB calculations as requested.

We have calculated the total luminescent quantum yield for Ln(III) emission according to established conventions in the literature. However, as the referee has kindly pointed out in this instance, it is mandatory that the employed transition specific quantum yield is used for CPB calculation with respect to detection window and possible CPL transition sign cancellation. This very last undesired feature is the reason behind the synthetic design of Eu:L1 to construct a bright (high molar extinction coefficient and quantum yield) CPL active (high g_{lum}) complex with broad single CPL transition to maximise detection window and true measurable CPB.

(Comment 3) Authors should introduce why to design the new complex Eu:L1 in manuscript.

Response to Reviewer 2 Comment 3

We apologise if we have been rather short in describing the underlying photophysical desire that led to the synthesis of Eu:L1. As detailed above (comment 2) we have now expanded on this point in the manuscript and also included an important new figure underpinning the need for the design of Eu:L1 in the SI.

Reviewer #3 (Remarks to the Author):

(Comment 1) This short, but ultra-technique communication entitled “Rapid handheld time-resolved circularly polarised luminescence (CPL) photography camera for life and material sciences” has been submitted to Nat. Comm. This reviewer carefully read this contribution and congratulate the authors for producing a valuable issue with a top-quality scientific level. However, this modest referee thinks that the degree of novelty and originality, compared with the previous publications of the same team: Chem. Commun., 2016, 52, 13349, Nat. Commun., 2020, 11, 1676 and Nat. Commun., 2022, 13, 553, does not reach the threshold to be published in Nat. Commun. A real-life example such as implementing the chiral Eu chromophore as a security ink, and using their technology to detect the CPL would be interesting and novel.

Response to Reviewer 3 Comment 1

We thank the reviewer for the encouraging comments and excitement regarding our work and we apologise if we appeared to be modest in emphasising the novelty and originality of our latest milestone achievement. Further to our responses above to Reviewer 1’s similar reservations we hope to assure the reviewer that indeed our latest CPL instrumentation development is on par with our previous instrumentation milestones warranting their publication as part 3 on our instrument trilogy.

More specifically, we would like to direct the reviewer to the two new additional figures in the main text (Figure 6 and 7 supported by Figure SI 23 and 24) demonstrating a proposed real life 'rudimentary' POC CSI security ink design and its underlying five-fold security feature to be instantly read out by our CPLP camera - ad hoc time-resolved EDCC based one-shot CPL photography via chromo-spatio-temporal-spectral-chiral discrimination. We hope that our combined reasoning to all comments raised by the reviewers and incorporation of new additional information to both main article text and SI (Figures 6 and 7 and Figure SI 1, 4, 8, 9, 23 and 24) manage to convince the reviewer that our camera system is ground-breaking, novel, has widespread applications in both life and material sciences and is indeed worthy of being published in Nature Communications.

(Comment 2) In table SI 1 there are few mistakes concerning the calculated CPB. If using the formula: $CPB = \xi_{abs} \phi_{em}(gem/2)$, the calculated CPB values obtained by this reviewer for Eu:L1, Eu:L2, Eu:L3, Eu:L4, Eu:L5 and Eu:L6 are not in agreement with the reported by the authors. The values obtained are half of those reported. Please check the numbers.

As per our comments to **Reviewer 2 Comment 2** we have already rectified this.

(Comment 3) In the SI, authors have mentioned that they used two standards for the quantum yields measurements. Please add the chemical name/formula of those standards. **AND (Comment 4)** The authors mention "Chiral HPLC analysis" in the main text and SI but they did not show any chromatogram neither in the main text nor in the SI. Please add the chromatograms of the Eu:L1.

We have now rectified both and included these as requested.

We hope that these comments, reasonings aided by corrections and modifications to the original text satisfy the reviewers.

Sincerely,

Robert Pal

Reviewer #1 (Remarks to the Author):

I changed my mind about their novelty as they were very well explained. I found that there are many technological progresses in CPL microspectroscopy. If papers in applied fields are also accepted by Nature Chem, I think this paper is well worthy of publication.

Reviewer #2 (Remarks to the Author):

This revision answered all my concerns. The quality of the paper is improved. I recommend the publication of the paper in its current form.

Reviewer #3 (Remarks to the Author):

After reading the changes that the authors have made I consider that the paper can be accepted in Nat. Comm.

Reviewer #4 (Remarks to the Author):

The manuscript entitled "Rapid handheld time-resolved circularly polarised luminescence (CPL) photography camera for life and material sciences" reported a solid-state small footprint CPL camera with no moving parts to facilitate ad hoc time-resolved enantioselective differential chiral contrast (EDCC) based one-shot CPL photography (CPLP).

I have carefully read the manuscript as well as the reviewer's comments. This is an interesting work, the demonstrations are well designed, and the results are solid. However, I agree with referees 1 and 3 that the novelty of this work is not sufficiently to meet the high standard of Nature Communications since similar results have been already reported.

Minor comment: I personally think that the introduction section is too long and it should be shortened.

Authors Response to Editor's Reviewers' Comments

We greatly appreciate the editors and reviewers' constructive comments and have attempted to address them in detail in the revised manuscript.

REVIEWERS' COMMENTS

Reviewer #1 (Remarks to the Author):

I changed my mind about their novelty as they were very well explained. I found that there are many technological progresses in CPL microspectroscopy. If papers in applied fields are also accepted by Nature Chem, I think this paper is well worthy of publication.

Thank you for the kind words, much appreciated.

Reviewer #2 (Remarks to the Author):

This revision answered all my concerns. The quality of the paper is improved. I recommend the publication of the paper in its current form.

Thank you for the kind words, much appreciated.

Reviewer #3 (Remarks to the Author):

After reading the changes that the authors have made I consider that the paper can be accepted in Nat. Comm.

Thank you for the kind words, really appreciated.

Reviewer #4 (Remarks to the Author):

The manuscript entitled "Rapid handheld time-resolved circularly polarised luminescence (CPL) photography camera for life and material sciences" reported a solid-state small footprint CPL camera with no moving parts to facilitate ad hoc time-resolved enantioselective differential chiral contrast (EDCC) based one-shot CPL photography (CPLP).

I have carefully read the manuscript as well as the reviewer's comments. This is an interesting work, the demonstrations are well designed, and the results are solid. However, I agree with referees 1 and 3 that the novelty of this work is not sufficiently to meet the high standard of Nature Communications since similar results have been already reported.

Thank you for your comments, we appreciate that both Reviewer 1 and 3 have changed their mind regarding the novelty and cutting-edge nature of our research article and we sincerely hope that this alongside our revised paper including new figures would also change the mind of Reviewer 4.

Minor comment: I personally think that the introduction section is too long and it should be shortened.

After considering this request, and reading the introduction again, we would like to respectfully disagree with this request. In order to allow the multidisciplinary readership to gain full appreciation and detailed explanation of our aims and objectives, the introduction has been carefully constructed to provide the best possible introduction, chronological overview of the underpinning technique CPL spectroscopy and microscopy. Therefore, we feel that if any section would be removed or truncated vital information would be lost leaving the reader with unanswered questions regarding CPL instrument development journey from its humble origins to this latest development.

Sincerely,

Robert Pal